# Adventitious carbon breaks symmetry in oxide contact electrification

Galien Grosjean[1,2✉], Markus Ostermann[3], Markus Sauer[4], Michael Hahn[5], Christian M. Pichler[3,6], Florian Fahrnberger[5], Felix Pertl[1], Daniel M. Balazs[1], Mason M. Link[7], Seong H. Kim[7], Devin L. Schrader[8], Adriana Blanco[9], Francisco Gracia[9], Nicolás Mujica[10] & Scott R. Waitukaitis[1✉]

Insulating oxides are among the most abundant solid materials in the universe[1–3]. Of the many ways in which they influence natural phenomena, perhaps the most consequential is their capacity to transfer electrical charge during contact[4–10]—which occurs even between samples of the same oxide—yet the symmetry-breaking parameter that causes this remains unidentified[11,12]. Here we show that adventitious carbonaceous molecules adsorbed from the environment are the symmetry-breaking factor in same-material oxide contact electrification (CE). We use acoustic levitation to measure charge exchange between a sphere and a plate composed of identical amorphous silicon dioxide ($SiO_2$). Although charging polarity is random for co-prepared samples, we control it with baking or plasma treatment. Observing the charge-exchange relaxation afterwards, we see dynamics over a timescale of hours and connect this directly to the presence of adventitious carbon with time-of-flight mass spectrometry, low-energy ion scattering and infrared spectroscopy. Going further, we confirm that adventitious carbon can even determine charge exchange among different oxides. Our results identify the symmetry-breaking parameter that causes insulating oxides to exchange charge in settings ranging from desert sands[4] to volcanic plumes[5,6], while simultaneously highlighting an overlooked factor in CE more broadly.

$SiO_2$ accounts for up to 60% of the Earth's crust, primarily in minerals such as olivine and pyroxene[1]. Extraterrestrial bodies, including the Moon[2], rocky planets[3], chondrules and asteroids[2], are derived from $SiO_2$ and other oxides ($Al_2O_3$, $MgO$ and so on). Given their ubiquity, these materials play important roles in many natural processes. Of particular consequence is the exchange of electrical charge, or CE, that occurs when two oxides touch. For instance, electrostatic forces on CE-charged particles are implicated in the long residence times and extraordinary distances of fine sands ($SiO_2$) transported during Saharan dust storms[4]. In volcanic plumes, CE creates displays of 'volcanic lightning'[5,6] that, beyond spectacle, may have provided an energy source to convert primordial molecules into amino acids[7]. On the Moon, Mars and asteroids, CE presents substantial challenges during space missions[8,9,13]. Within stellar disks, attractive forces from CE are crucial for the early stages of rocky planet growth[10], helping large objects form before gas drag causes them to spiral into their stars[14,15].

Despite this universal relevance, we do not know why CE occurs in most situations, including the restricted case of insulator oxides. Furthermore, CE does not just occur between different oxides (for example, $SiO_2$ and $Al_2O_3$) but also between samples of the same oxide (for example, $SiO_2$ and $SiO_2$). In the scenarios mentioned above, this 'same-material' CE is at play, despite any obvious symmetry-breaking parameter. Experiments show that work-function differences drive electron transfer between metals[16–18], but no related mechanism has been demonstrated for oxides, different or same material. Different-oxide CE may correlate with acid/base properties, yet this again fails to explain same-material CE (refs. 19–21). Other experiments show that adding even a single monolayer of artificial molecular adsorbates can alter CE behaviour. For example, a fused silica (amorphous $SiO_2$) surface dressed with (γ-aminopropyl)dimethylethoxysilane charges positively to a 'natural' surface[22] (that is, clean and stored in air). Similarly, soda–lime glass dressed with trimethylchlorosilane charges negatively to 'natural' soda lime[23]. This latter result[23] is one of many that have pointed towards adsorbed water as an important factor in CE, with oxides and beyond[24–29], although a definitive connection has remained elusive. Considering these factors, we might speculate that the symmetry-breaking factor responsible for same-material oxide CE resides in the 'natural' adsorbates that adhere to surfaces from their environment. Yet this raises questions. Do 'natural' adsorbates alone affect oxide CE at all? Which ones matter? And how can two samples in the same environmental conditions acquire differences in these adsorbates to break symmetry?

[1]Institute of Science and Technology Austria, Klosterneuburg, Austria. [2]Departament de Física, Universitat Autònoma de Barcelona, Bellaterra, Spain. [3]Centre for Electrochemical Surface Technology, Wiener Neustadt, Austria. [4]Analytical Instrumentation Center, Technische Universität Wien, Vienna, Austria. [5]Institute for Chemical Technology and Analytics, Technische Universität Wien, Vienna, Austria. [6]Institute for Applied Physics, Technische Universität Wien, Vienna, Austria. [7]Department of Chemical Engineering and Materials Research Institute, Pennsylvania State University, University Park, PA, USA. [8]Astromaterials Research and Exploration Science (ARES) Division, XI3 Research Office, NASA Johnson Space Center, Houston, TX, USA. [9]Departamento de Ingeniería Química, Biotecnología y Materiales, Facultad de Ciencias Físicas y Matemáticas, Universidad de Chile, Santiago, Chile. [10]Departamento de Física, Facultad de Ciencias Físicas y Matemáticas, Universidad de Chile, Santiago, Chile. ✉e-mail: galien.grosjean@uab.cat; scott.waitukaitis@ista.ac.at

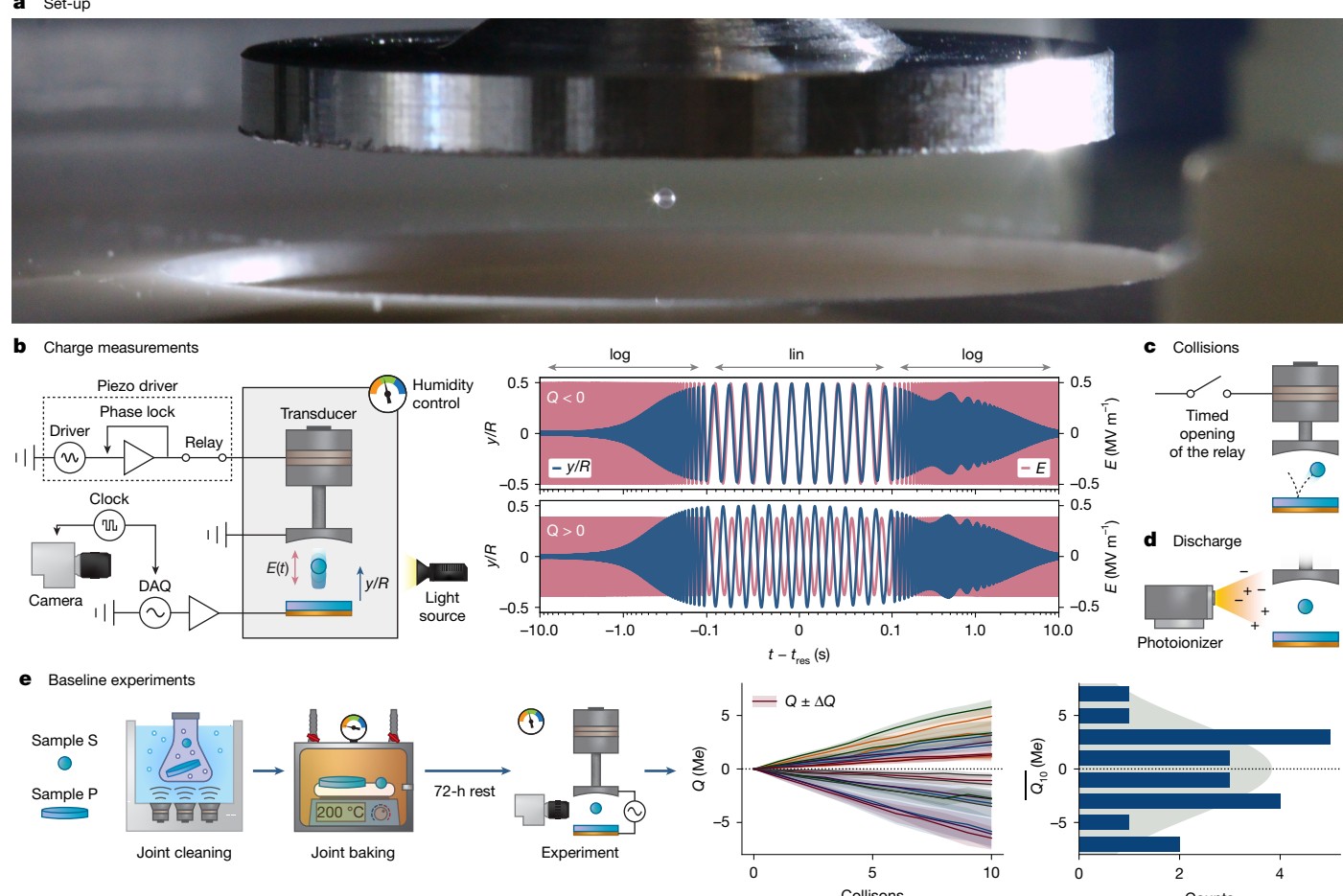

**Fig. 1 | Experimental set-up and baseline behaviour of nominally identical silica samples. a**, We levitate a 500-µm silica sphere in an acoustic trap above a silica target plate. **b**, Charge is measured by applying a frequency-swept electric field, $E(t)$, and extracting the trajectory of the sphere, $y(t)$, using a high-speed camera. The amplitude and phase of $y(t)$ at resonance depend, respectively, on the magnitude and sign of the particle's charge, $Q$. **c**, Timed interruptions of the acoustic field trigger charge-exchanging collisions between the sphere and the plate, in which the sphere falls, bounces off the plate and is then 'caught' again. **d**, The entire system can be discharged using an X-ray photoionizer. **e**, Every sphere charges with a systematic sign and magnitude against its partner plate, but the slopes for an ensemble of sphere/plate pairs are spread randomly about zero.

We explore these questions with the experiment shown in Fig. 1a–d. We use acoustic levitation to suspend a small ($d \approx 500$ µm) spherical particle above a plate (Fig. 1a), both made from high-purity fused silica (Methods). We measure the charge of the sphere by applying an AC electric field sweep across the acoustic cavity (Fig. 1b) and fitting its trajectory to the equation of motion (Methods). To cause a collision, we briefly (roughly 25 ms) interrupt the acoustic field, letting the sphere fall and bounce off the plate (Fig. 1c). At the apex of its rebound, we reinitiate to 'catch' the sphere. Supplementary Video 1 further illustrates how the experiments work. We ensure that spheres/plates begin with zero charge through photoionization of the surrounding air (Fig. 1d and Extended Data Fig. 1). Spheres/plates are subjected to a standardized cleaning protocol: 30 min sonication in acetone, 30 min sonication in methanol, 30 min sonication in ultrapure water, 2 h baking at 200 °C and then storage for >72 h in the experimental chamber. Humidity (30 ± 1% relative humidity) and temperature (25 ± 1 °C) are controlled.

Figure 1e shows the charge acquired over sequential bounces for an ensemble of sphere/plate pairs (see Methods for details on protocol and error bands). Each sphere charges with a definite sign and constant slope against its partner plate, yet for many pairs, the slopes spread randomly about zero. This behaviour—a systematic difference between any two samples but randomized across the ensemble—can be considered the baseline for what is to follow[30]. Paradoxically, it indicates that each SiO$_2$ sample behaves as a different material in relation to CE.

To test whether atmospheric adsorbates underlie this behaviour, we alter samples not by addition of artificial molecules[22,23] but instead by subtraction of those naturally present. We do this with two common treatments: exposure to low-power plasma or mild baking. Both are widely used for removal of adsorbates, for example, in ultrahigh vacuum (UHV) or cleanrooms[31]. The procedure is shown in Fig. 2a. Two samples are prepared jointly with the standard protocol and their 'baseline' CE is measured. Then, one sample (S or P) is retrieved and either plasma treated (5 min) or baked (typically 2 h at 200 °C). The CE measurement is repeated immediately thereafter.

Figure 2b shows results for the case of plasma. The blue curves correspond to the baseline charging of two spheres against their partner plates. In both instances, CE happens to be positive at approximately $10^5$ $e$ per collision. After plasma treating a sphere, it charges negatively (about $-10^5$ $e$ per collision). Conversely, treating a plate results in its partner sphere charging more positively (about $5 \times 10^5$ $e$ per collision). Baking has the same effect: baking a sphere causes it to charge more negatively, whereas baking a plate causes its partner sphere to charge more positively (Fig. 2c). Temperature and duration matter: baking at 300 °C for 2 h almost always results in negative charging; baking

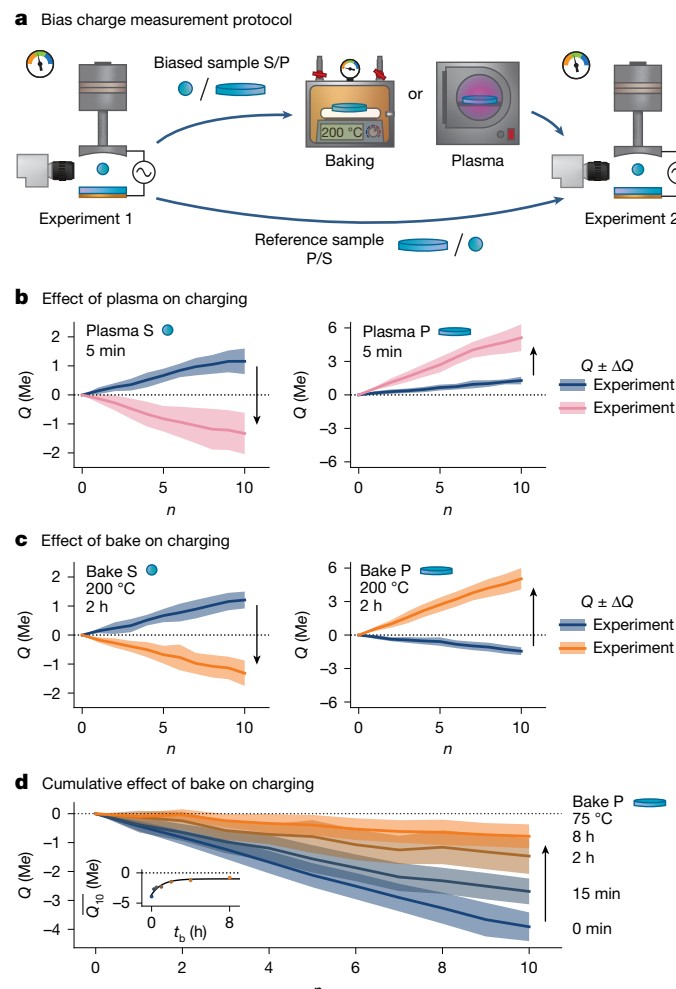

**a** Bias charge measurement protocol

Biased sample S/P

Baking | or | Plasma

Experiment 1

Experiment 2

Reference sample P/S

**b** Effect of plasma on charging

Plasma S
5 min

Plasma P
5 min

$Q \pm \Delta Q$
— Experiment 1
— Experiment 2

**c** Effect of bake on charging

Bake S
200 °C
2 h

Bake P
200 °C
2 h

$Q \pm \Delta Q$
— Experiment 1
— Experiment 2

**d** Cumulative effect of bake on charging

Bake P
75 °C

8 h
2 h

15 min

0 min

**Fig. 2 | Controlling charging behaviour by removing 'natural' adsorbates.** **a**, We introduce a bias by exposing one sample, plate (P) or sphere (S) to either heat or mild plasma, with the intention of removing naturally occurring adsorbates. **b**, Spheres treated with plasma always charge negatively to their partner plate, regardless of how they charged before. Conversely, if the plate is treated with plasma, its partner sphere will charge more positively. **c**, Baking affects the samples in a similar way: baked samples always charge more negatively, to the point of polarity reversal if the temperature/duration of the bake is sufficiently high/long. **d**, Even a gentle bake of 75 °C for 15 min has a measurable effect, which is cumulative if done successively.

at 200 °C for 2 h usually does; but even 75 °C for 15 min has an effect, which is cumulative if done successively (Fig. 2d).

These data contradict widespread arguments implicating adsorbed water as the symmetry breaker in oxide CE. Generally, it is argued that a hydrophilic surface charges positively to a hydrophobic one by means of donation of OH⁻ (refs. 23,25). Plasma and baking make samples more hydrophilic and adsorbed water reaches equilibrium quickly owing to its abundance in air[32,33]. Yet we observe that these surfaces charge negatively, not positively. Figure 2 therefore offers provisional insight into our first question; naturally occurring adsorbates do indeed matter in same-material oxide CE but it is not clear that water is the symmetry breaker.

To search for other candidates, we perform time-of-flight secondary ion mass spectrometry (ToF-SIMS) on a silica plate after our standard cleaning protocol (Fig. 3a). Far from just $SiO_2$ and $H_2O$, the surface hosts a cocktail of molecular and atomic species, with adventitious carbon—that is, naturally occurring carbonaceous molecules from the environment—as the marquee ingredient. Detected species include

small moieties such as $CH_3$ ($m/z$ = 15) but also larger ones such as $C_3H_6$, $C_4H_{10}$ and $C_6H_{10}$ ($m/z \approx$ 42, 58 and 82, respectively). We use spatially resolved ToF-SIMS to image two of these, $C_3H_6$ and $C_4H_{10}$, which reveals extensive coverage (Fig. 3b). If we plasma treat or bake, these species are greatly reduced (see also inset to Fig. 3a and Extended Data Fig. 2). If we remove the sample from UHV, store it in air for several hours and image again, some return. In contrast to water, which readsorbs to an air-exposed[32,33] or even UHV-exposed[34] surface essentially instantly, these data suggest hours-scale dynamics in the readsorption of adventitious carbon. This allows us to make a prediction about the CE behaviour after baking or plasma—if it is connected to adventitious carbon, it should slowly change with time.

We test this prediction by measuring the charging rate (average charge after ten collisions, $\overline{Q_{10}}$) versus time after treatment. Extended Data Fig. 3 confirms that, after standard cleaning (and the 72-h wait), the charging rate between two samples is stable. Figure 3c shows the evolution immediately after plasma. The two data points before $t$ = 0 h correspond to the initial stable rate. Immediately afterwards, the rate becomes strongly negative, consistent with Fig. 2b. However, thereafter, it evolves over a scale of hours, trending towards its pretreatment value but stabilizing at a new one. As we show in Extended Data Fig. 4, charge-exchange relaxation after baking occurs in the same way. The timescale varies from run to run but is typically around 10 h.

Notably, the readsorption of adventitious carbon exhibits nearly identical post-treatment dynamics. We ascertain this using low-energy ion scattering spectroscopy (LEIS) to monitor the atomic composition of the outermost atomic/molecular layer versus atmospheric storage time (Fig. 3d). Baseline LEIS signals before plasma ($t$ < 0 s) corroborate the presence of adventitious species, with both C and H present (Methods and Extended Data Fig. 5). Immediately after plasma, the Si and O signals increase sharply, whereas C plummets. This confirms the removal of the carbon layer and exposure of the bulk. As a function of storage time, all signals slowly move towards their baseline values but, like the charging data, settle at new ones. Using Fourier transform infrared (FTIR) spectroscopy (Methods), we observe largely identical trends when monitoring $CH_x$ stretching modes before and after baking (Extended Data Fig. 6). Crucially, the timescale for the formation of a full carbon layer on the surface, as judged by the evolution of both the LEIS Si/O/C and FTIR $CH_x$ signals, is also around 10 h—as observed in the charge-exchange measurements.

The fact that carbon readsorption and charging evolve at a similar rate can be considered a 'smoking gun' in the context of CE. In other domains, however, adventitious carbon is so ubiquitously dealt with that its presence and influence might be taken for granted. For example, X-ray photoelectron spectroscopy (XPS) studies routinely use adventitious carbon as a binding energy standard—precisely because it is present on all surfaces[35,36]. XPS experiments show that adventitious carbon accumulates indefinitely, for which one exceptionally long study found no bound even after 7 months[37]. Outside vacuum, FTIR spectroscopy has connected adventitious carbon to hydrophobic recovery after plasma treatment[38], for which the contact angle evolution occurs on a similar timescale (see Extended Data Figs. 6 and 7, in which we have confirmed this ourselves). Such experiments reveal that the relevant molecules are: (1) present in air at the parts per trillion scale[38]; (2) susceptible to environmental fluctuations; and (3) not in equilibrium with the surface[37]. Hence, the precise cocktails on two objects in the same conditions need not be the same—they will depend on the histories of the objects. Our observations reiterate this history dependence: (1) charging rates do not revert to their pretreatment values after plasma/baking (Fig. 3c); (2) LEIS Si/O/C and FTIR $CH_x$ signals similarly do not revert (Fig. 3d and Extended Data Fig. 6); and (3) relaxation timescales vary from run to run (Extended Data Fig. 4). Returning to our third question: because of this history-dependent, out-of-equilibrium nature, differences in carbon coverage are the rule not the exception.

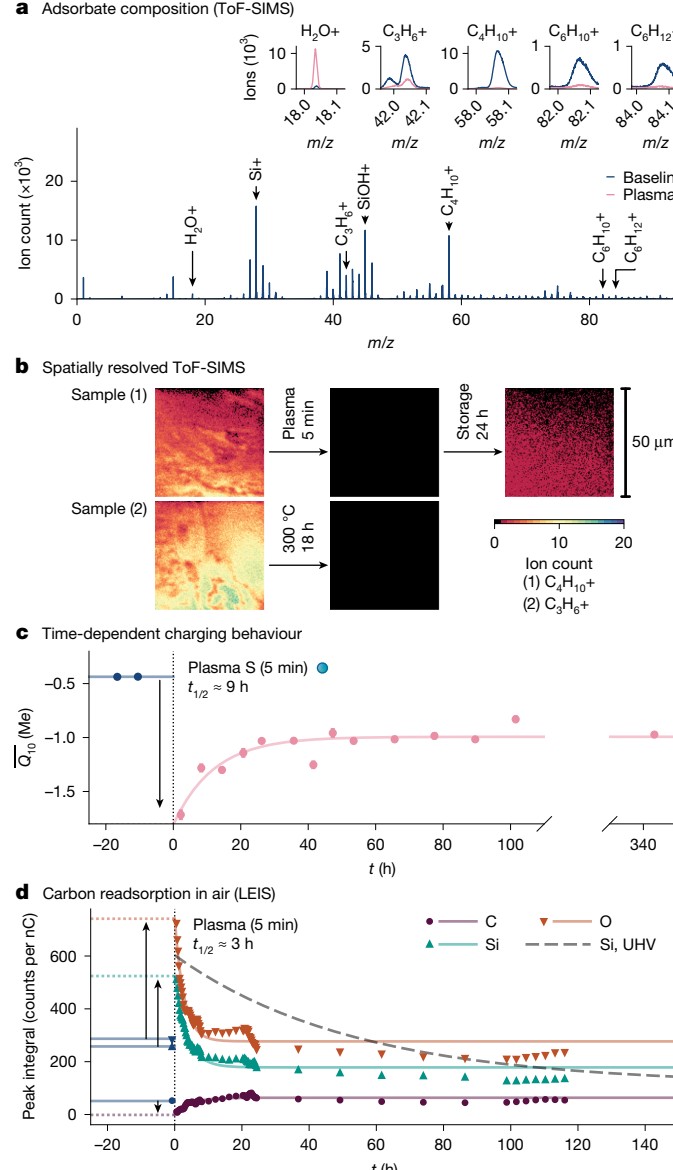

**Fig. 3 | Charging behaviour correlates with the presence of adventitious carbon. a**, Spectra from ToF-SIMS show a cocktail of adsorbed species, especially carbon-based moieties. After plasma treatment (inset plots) or bake (Extended Data Fig. 2), peaks corresponding to these are greatly reduced. The $H_2O$ peak increases, highlighting the quick readsorption and equilibration of water, whereas peaks corresponding to the $SiO_2$ matrix are broadly unchanged. **b**, Spatially resolved ToF-SIMS shows the spread of carbon species about the surface. These species are removed by plasma or baking but then readsorb over time. **c**, Although a measured charging rate is constant after our standard cleaning and 72-h storage, it evolves over time directly after plasma/baking. The relaxation half-life is typically around 10 h. **d**, We measure the time-dependent atomic composition of a surface's outermost atomic layer using LEIS, exposing it to air between measurements. The signal for C is reduced to virtually zero immediately after plasma, then comes back over the same order of time as the charging relaxation. The H signal follows a similar trend, although with an offset at $t = 0$ h owing to the fast equilibration of silanols and surface water. The Si and O increase immediately after plasma as the bulk is exposed but then decrease as the carbon layer develops. We observe the same trends if we keep a sample in UHV, although with a much longer timescale (about 36 h, dashed line).

These results naturally raise a broader question: do adsorbed carbonaceous molecules also influence CE between different oxides? We imagine three scenarios. In the first, a 'strong' carbon effect controls all oxide charging, same or different material. In a second 'intermediate'

scenario, both the carbon and the material properties matter. In a third 'weak' scenario, material effects always dominate. To investigate these possibilities, we assemble samples of several different oxides: alumina ($Al_2O_3$), spinel ($MgAl_2O_4$), silica ($SiO_2$) and zirconia ($ZrO_2$, stabilized with $Y_2O_3$). These vary by solid structure (amorphous, monocrystalline and polycrystalline), compositional homo/heterogeneity (simple oxides, complex oxides and oxides doped with impurities) and orders of magnitude in roughness (from about 1 nm to >200 nm). The materials and their properties are listed in Fig. 4a. We first determine the baseline behaviour of 'natural' inter-oxide CE after standard cleaning. The CE outcomes for all material combinations are plotted in matrix form in Fig. 4b, in which the colour indicates the charging rate $\overline{Q_{10}}$ corresponding to either the column ((1)–(5)) or the line material ((6)). Despite marked variability in roughness, crystallinity and so on, these samples form a coherent triboelectric series, that is, a transitive ordering on the basis of the sign of charge acquired. For instance, alumina charges positively against all others, placing it atop the series, whereas spinel charges negatively to alumina but positively to the rest and so on.

These data are consistent with previous reports[19–21] and seemingly support the intermediate or weak scenarios. To differentiate between the two, we perform further experiments with carbon-removing treatments. After each baseline experiment, we retrieve whichever sample charged positively, bake it and perform a second measurement. In every case, removal of carbon reverses the sign of charging (Fig. 4c). This is akin to inverting the triboelectric series (Fig. 4d). As we show in Extended Data Fig. 8, we cause the same sign flips with plasma and the effect extends to all oxides we have tried (MgO, sapphire, borosilicate glass, soda–lime glass). These data support the intermediate scenario. If carbon asymmetry is minimal, charging is influenced by the underlying material. But if one surface is largely stripped of carbon, the material effect can be overridden.

The literature is replete with allusions to 'surface contamination' as a confounding factor to be considered in CE, both with oxides and generally, sometimes even calling out 'organic contaminants', 'hydrocarbons' or 'adventitious carbon' by name[11,28,39–43]. However, we are not aware of studies specifically aimed at understanding its role. It has long been considered as 'noise' and not 'signal'. Here we come to an alternative interpretation: the difference in the adventitious carbon on two same-material oxide surfaces is the symmetry-breaking factor that underlies why they exchange charge. As we show in Extended Data Fig. 9, iteratively removing carbon from both samples of a plate/sphere pair leads to the suppression of CE altogether. For different-oxide CE, the 'noise' interpretation is more appropriate. UHV experiments to study CE in the 'absence' of adventitious carbon would be valuable, but even there it reaccumulates (see Extended Data Fig. 10). Moreover, the natural phenomena that make oxide CE consequential occur in conditions far less pristine than UHV.

We restrict our conclusions about adventitious carbon to CE among insulating oxides but we would not be surprised if they extend more broadly. For the oxide CE mechanism at the atomic/molecular level, our experiments do not permit a conclusion but do bring to light a largely overlooked factor—it is sensitive to trace carbonaceous adsorbates from the environment. Proposed mechanisms should be compatible with this fact. It is well known that carbon adsorption alters the work function of a metal[44,45]. Hence, if some 'work-function' mechanism occurs with oxides, differences in carbon coverage could matter. Theoretical work argues[46] that adsorbates can alter and potentially flip the sign of flexoelectric coefficients. Although we are not aware of experimental evidence for this, if it is confirmed, then mechanisms based on flexoelectricity could be plausible[46–48]. For the role of water in oxide CE (refs. 25–28), our data show that it is not as simple as previously imagined. We foresee mechanisms involving both carbon and water, for example, in which differences in carbon break symmetry whereas water enhances charge mobility[49]. Although it is clear that adventitious carbon matters critically, these issues remain unresolved.

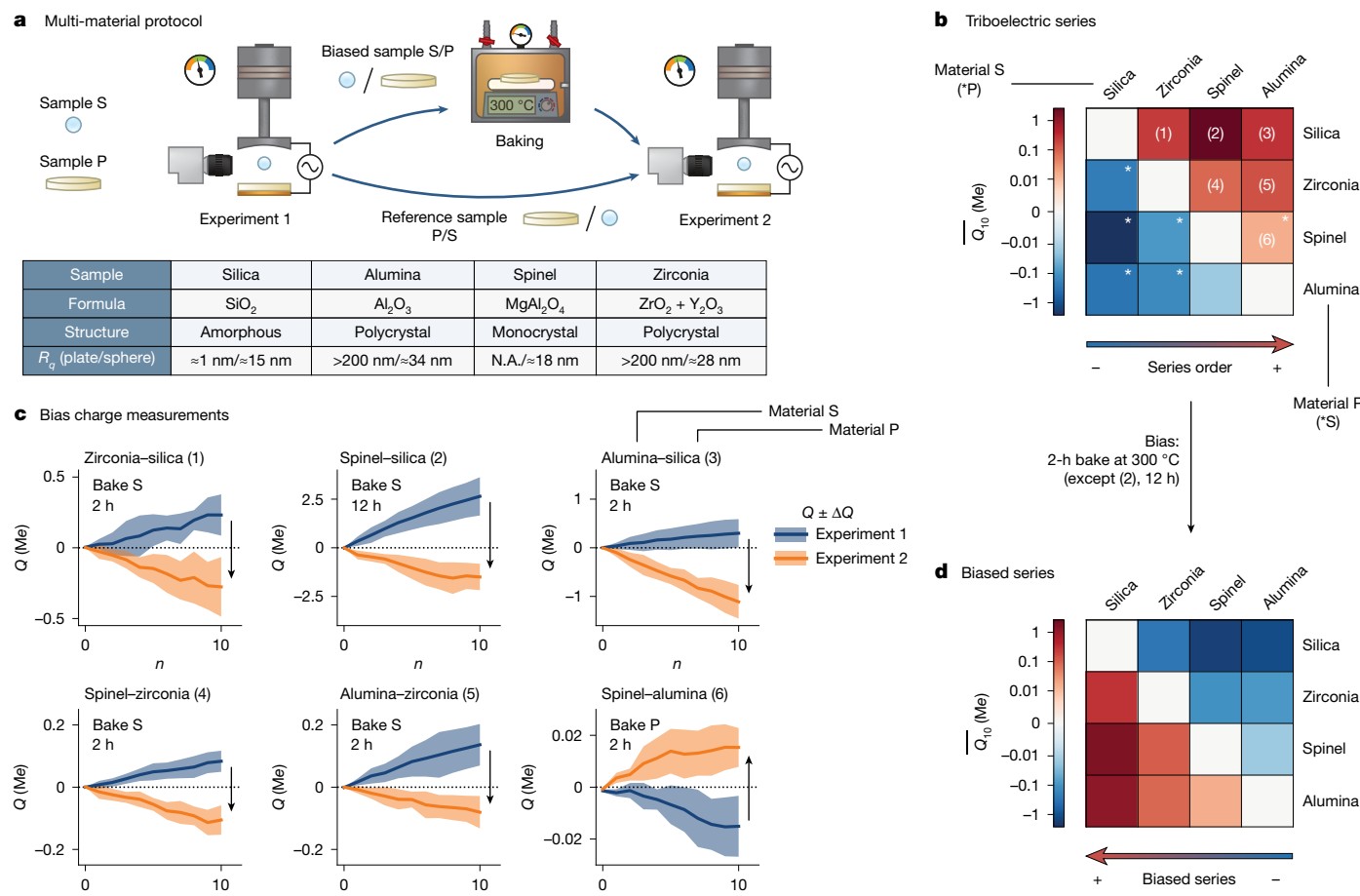

**a** Multi-material protocol

| Sample | Silica | Alumina | Spinel | Zirconia |
|---|---|---|---|---|
| Formula | $SiO_2$ | $Al_2O_3$ | $MgAl_2O_4$ | $ZrO_2 + Y_2O_3$ |
| Structure | Amorphous | Polycrystal | Monocrystal | Polycrystal |
| $R_q$ (plate/sphere) | ≈1 nm/≈15 nm | >200 nm/≈34 nm | N.A./≈18 nm | >200 nm/≈28 nm |

**b** Triboelectric series

**c** Bias charge measurements

**d** Biased series

**Fig. 4 | Control of charging behaviour with different-material samples.**
**a**, Samples of four different oxides, with different composition, surface roughness and crystalline structure, are tested against each other. **b**, When measuring every pair combination with just our standard cleaning protocol, a perfect triboelectric series forms with alumina ($Al_2O_3$) at the top and silica ($SiO_2$) at the bottom. **c**, For every pair combination, we bake the positively charging sample (sphere or plate) to reverse the charging polarity. **d**, The result is an 'inverted' triboelectric series with $SiO_2$ at the top and $Al_2O_3$ at the bottom. These data illustrate that the 'carbon effect' competes with an underlying material effect. The latter is stronger if full carbon layers are present, whereas the former dominates if one surface is largely absent from carbon.

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

## Methods

### Silica samples

The main samples used throughout the paper are made of amorphous $SiO_2$, more specifically, high-purity synthetic ultraviolet-fused silica (Heraeus Spectrosil 2000). The spheres, produced by Sandoz Fils SA, have a diameter of $500 \pm 0.7$ μm (measured over ten samples) and a nominal deviation from sphericity of 0.625 μm (ANSI/ABMA 10A-2001 grade 25). The plates, produced by UQG Optics Ltd. (ref. WFS-252), are discs 25 mm in diameter and 2 mm thick. Using an atomic force microscope (Park Systems NX20), we determined the root mean square (r.m.s.) roughness $R_q$ of the spheres to be about 5 nm and the roughness of the plates to be about 1 nm. We measured no difference in roughness after exposure to plasma.

### Other samples

The other oxide samples are listed below, starting with the materials used in the triboelectric series (Fig. 4). Composition percentage is by weight unless otherwise specified. The nominal particle diameters are indicated below; however, the precise diameter is determined for each individual particle with the experimental camera. The r.m.s. roughness of all samples was measured using atomic force microscopy (AFM; Park Systems NX20) with first-order and second-order image flattening for the plates and spheres, respectively. Crystalline structure was determined using powder X-ray diffraction (XRD; Bruker D8 ADVANCE) for the flat samples, whereas the spherical samples were characterized using a combination of transmission X-ray diffraction (Xenocs Xeuss 3.0 HR), Laue backscattering (Photonic Science Laue crystal orientation system) and single-crystal X-ray diffraction (Rigaku Synergy-S diffractometer).

Alumina spheres (Goodfellow Cambridge Ltd., ref. AL606810) are composed of 99.99% $Al_2O_3$ (CAS number 1344-28-1) and have a nominal diameter of $500 \pm 2.5$ μm (ANSI/ABMA 10A-2001 grade 25) and a measured r.m.s. roughness of about 25 nm.

Alumina plates (Almath Crucibles Ltd.) are composed of 99.8% $Al_2O_3$ and have a diameter of 25 mm, a thickness of 2.5 mm and an r.m.s. roughness of at least 200 nm.

Spinel spheres (Sandoz Fils SA) are composed of 99.99% $MgAl_2O_4$ and have a measured diameter of $504 \pm 0.5$ μm and an r.m.s. roughness of about 18 nm.

Zirconia spheres (Cospheric LLC, ref. YSZMS-6.05 425–500 μm) are composed of 8Y-FSZ (zirconia stabilized with 8 mol% yttria), which is about 87% $ZrO_2$ and 13% $Y_2O_3$ (CAS number 114168-16-0). Their diameters range between 425 and 500 μm and we determined their r.m.s. roughness to be roughly 14 nm.

Zirconia plates (Almath Crucibles Ltd.) are composed of 8Y-FSZ (zirconia stabilized with 8 mol% yttria), which is about 87% $ZrO_2$ and 13% $Y_2O_3$, and have a diameter of 25 mm, a thickness of 2.5 mm and an r.m.s. roughness larger than 200 nm.

Other samples, which were tested (Extended Data Fig. 8) but not used in the triboelectric series measurements (Fig. 4), are listed below.

Magnesia (MgO) plates (Almath Crucibles Ltd.) are composed of 98.3% MgO and 1.2% $Y_2O_3$ and have a diameter of 25 mm and a thickness of 2.5 mm.

Sapphire spheres (Sandoz Fils SA) are composed of 99.99% $Al_2O_3$ and have a nominal diameter of $500 \pm 2.5$ μm (ANSI/ABMA 10A-2001 grade 25).

Sapphire plates (UQG Optics Ltd., refs. WSR-252 and WSC-252) are composed of high-purity (99.998% and 99.996%, respectively) $Al_2O_3$ and have a diameter of 25 mm, a thickness of 2 mm and an r.m.s. roughness of about 0.5 nm.

Soda–lime glass spheres (Goodfellow Advanced Materials, ref. SI80-SP-000150) are composed of 70% $SiO_2$, 15% $Na_2O$, 10% CaO and small quantities of MgO, $B_2O_3$ and $Al_2O_3$. They have a nominal diameter of $500 \pm 70$ μm.

Borosilicate glass spheres (Sandoz Fils SA) are composed of 81% $SiO_2$, 13% $B_2O_3$, 4% $Na_2O/K_2O$ and 2% $Al_2O_3$ and have a measured diameter of $498 \pm 0.3$ μm.

Borosilicate glass plates (UQG Optics Ltd., ref. WBO-252) are composed of borosilicate 3.3 glass (Schott Borofloat 33), which is about 81% $SiO_2$, 13% $B_2O_3$, 4% $Na_2O/K_2O$ and 2% $Al_2O_3$, and have a diameter of 25 mm, a thickness of 2 mm and an r.m.s. roughness of about 1 nm.

### Sample preparation

The protocol for sample preparation is the same for all materials. Spheres and plates are cleaned jointly. After being washed with a mild soap and water, they undergo 30 min of sonication successively in acetone (Thermo Scientific Chemicals, >99.5%), methanol (Honeywell Riedel-de Haën CHROMASOLV, >99.9%) and ultrapure water (Merck Millipore Milli-Q), with abundant rinsing between steps using ultrapure water. The samples are then baked for 2 h on a hotplate at 200 °C enclosed in a dedicated box and then immediately moved to the experimental chamber, in which they are stored at 30% relative humidity for at least 72 h before measurements. Glazed alumina crucible boats are used for the baking, plasma treatment, storage and transport of the samples. Samples can be biased in two ways: exposure to plasma or baking. Plasma-treated samples are exposed to an air plasma at 500 mTorr (66.7 Pa) for 5 min in a Harrick PDC-002 set to the lowest power setting. Baked samples are placed on a hotplate for 2 h at 200 °C (unless otherwise specified). Charge measurements are performed immediately after bake/plasma.

### Charge measurements

All charge measurements were performed in the closed experimental chamber at 30% relative humidity. The acoustic trap consists of a Langevin transducer (Micromechatronics, Inc., STC-4SH-3540) fitted with a custom step-shaped horn terminated by a disc-shaped radiator with a spherical cut. To complete the acoustic cavity, the plate sample, which doubles as a reflecting plate, is placed on a tri-axis precision stage below the transducer. The distance between the plate and the lower point at the centre of the horn is tuned to precisely half of the wavelength of the ultrasound emitted by the horn, creating a standing wave. For a typical driving frequency of about 40 kHz, this corresponds to a distance of about 4 mm. The transducer is driven by a computer-controlled ultrasonic generator (PiezoDrive PDUS210), which uses a phase control loop to maintain the transducer in the same regime over long periods of time, at a frequency slightly above the impedance minimum corresponding to the first longitudinal vibration mode of the transducer. A relay is used to trigger brief, timed interruptions of the acoustic field, which allows performing collisions between the levitated particle and the target plate. A motorized stage allows us to use new contact locations on the plate for each collision.

Below the target plate, a copper electrode connected to the output of a high-voltage amplifier (UltraVolt, Inc., 5HVA24-BP1-F-25PPM) is used to apply electric fields across the acoustic cavity. Combined with the horn, which is electrically connected to ground, the electrode approximately forms a plate capacitor.

To determine the charge of the particle, we expose it to a rising frequency electric-field sweep centred on the natural frequency of the acoustic trap, typically about 50 Hz. The sweep is generated by a data acquisition system (National Instruments USB-6361, X Series DAQ device). The resulting oscillation of the particle is recorded using a high-speed camera (Phantom VEO 640L) and tracked to obtain the position of the particle as a function of time, $y(t)$. The camera and DAQ share a common clock so that the precise value of the electric field as a function of time, $E(t)$, is also known. Newton's second law projected along the vertical direction can be written

$$\ddot{y} = -g - a\sin 2ky - 2\beta_0 \dot{y} - 2\beta_1|\dot{y}|y + QE(t)/m$$

in which $a$ and $k$ are the acoustic amplitude and wavenumber, $\beta_0$ and $\beta_1$ are the linear and quadratic damping coefficients, and $Q$ and $m$ are the charge and mass of the particle. We calculate numerically the first and second derivatives of the particle position $\dot{y}$ and $\ddot{y}$ and determine $k$ and $m$ from the transducer frequency and the radius and density of the particle. This leaves us with four unknowns that can be determined by means of a fit: $a$, $\beta_0$, $\beta_1$ and $Q$. Both estimates from the fit residuals and successive measurements on the same charged sphere (repeated for several samples) point towards a measurement uncertainty on the order 1%, with a minimum uncertainty of about 500–1,000 $e$ (or about 0.1 fC) for spheres of relatively low charge ($\lesssim 10^5 e$). An X-ray photoionizer (Hamamatsu L12645, 10 keV), which sits inside the chamber but is not directed towards the samples, is used to discharge the sphere and plate before measurements. To determine $\overline{Q_{10}}$, we discharge the system, measure the initial charge $Q \approx 0\ e$ and then perform ten collisions/charge measurements. We then discharge the system again before performing another ten measurements. The data shown are typically the average and standard deviation calculated from ten such series of ten collisions.

## LEIS measurements

LEIS measurements were executed using an IONTOF Qtac$^{100}$ high-sensitivity spectrometer. The samples were fixed on a sample holder using double-sided copper tape. An oxygen-plasma gun within the UHV system was used to clean corresponding samples before measurement. $^4$He$^+$ were used at 3 keV at an incident angle of 0° and a scattering angle of 145°. To improve sensitivity levels, a time-of-flight mass filter was used. As well as the LEIS spectrum, a signal depending on sputtered H$^+$ was extracted using the recorded time-of-flight histogram. Effective measurement currents used for normalization of the measurement data were in the range 790–980 pA. The measurement area was set to $1,000 \times 1,000\ \mu m^2$ per measurement spot. A charge-compensation filament was used during LEIS measurements. We carried out data analysis using SurfaceLab 7 (IONTOF), Casa XPS (Casa Software Ltd.) and OriginPro (OriginLab).

## ToF-SIMS measurements

A time-of-flight secondary ion mass spectrometer (TOF.SIMS 5 instrument from IONTOF) was used to study adsorbates on a SiO$_2$ matrix. The experiment included 2D measurements on two specimens, first in the original state, then after one specimen (sample 1) had undergone a plasma-treatment procedure for 5 min and the other (sample 2) had been heated to 300 °C for 16 h to desorb species from the surface. After these first two measurements, sample 1 was left in a laminar flow hood overnight for 16 h to study readsorption. In each case, measurements were performed at two different randomly chosen locations on each specimen under the same conditions. No notable differences were observed between locations. A focused 25-keV Bi$^{3+}$ primary ion beam was used to generate secondary ions within a field of view of $500 \times 500\ \mu m^2$. All measurements were conducted with a pulse length of 17 ns and $512 \times 512$ pixels in random rendering. Bi$^{3+}$ was chosen over Bi$^+$ to obtain more information about higher-molecular-weight species (less fragmentation). Measurements were performed in positive polarity in high current bunched mode for a mass resolution of about 1,000–2,000, depending on species, and with currents of 0.8–0.9 pA (16–17 nA in DC). Considering the dose (approximately $8.8 \times 10^{11}$ ions cm$^{-2}$), these measurements stay within the static SIMS limit. The mass spectra were calibrated using the species CH$^+$, CH$_2^+$, CH$_3^+$, C$_2$H$_3^+$, C$_2$H$_5^+$, C$_4$H$_{10}^+$ and C$_6$H$_{12}^+$. The software used was SurfaceLab 7, version 7.1.130060.

## FTIR measurements

Diffuse reflectance infrared Fourier transform spectroscopy measurements were performed on a Thermo Fisher Scientific Nicolet iS50 spectrometer equipped with a liquid-nitrogen-cooled MCT detector and a PIKE Technologies diffuse reflectance cell, with controlled temperature, gas flow and KBr windows. Spectra were recorded in the range 600–4,000 cm$^{-1}$, with a resolution of 4 cm$^{-1}$ and 32 scans per spectrum. The main samples used were composite ZrO$_2$:SiO$_2$ particles with a diameter of $172 \pm 7\ \mu m$ (Glen Mills, Inc.), composed of 66% ZrO$_2$ and 29% SiO$_2$ (in weight) and an r.m.s. roughness $R_q \approx 60$ nm. Pure SiO$_2$ particles (Sigma-Adrich) were also used, sieved to a diameter of about 150 µm. We focused on the wavelength region 2,800–3,000 cm$^{-1}$, in which the vibrations of methylene (CH$_2$) and methyl (CH$_3$) groups were monitored. By deconvolution, we calculated the characteristic integrated absorbance of three peaks that correspond to the symmetric (sym) and asymmetric (asym) stretching vibrations of CH$_2$ (sym) at 2,850 cm$^{-1}$, CH$_2$ (asym) at 2,925 cm$^{-1}$ and CH$_3$ (asym) at 2,965 cm$^{-1}$.

## Data availability

Raw data are available from the corresponding authors on request. Source data for the figures are provided with this paper.

## Code availability

The scripts used to analyse the data in this study are available from the corresponding authors on request.

**Acknowledgements** This project has received support from the European Research Council (ERC) under the European Union's Horizon 2020 research and innovation programme (grant agreement no. 949120) and from the Marie Skłodowska-Curie programme (grant agreement no. 754411). We acknowledge the state of Lower Austria and the European Regional Development Fund under grant no. WST3-F-542638/004-2021. N.M. acknowledges support from grant Fondecyt 1221597. G.G. is a Serra Húnter fellow. This research was supported by the Scientific Service Units of the Institute of Science and Technology Austria through resources provided by the Miba Machine Shop, Nanofabrication Facility, Scientific Computing facility and Lab Support Facility. We thank the Modic group for the use of the Laue camera, T. Zauner for the photography of the experimental set-up and R. Möller for insightful discussions.

**Author contributions** G.G. conceived and performed all CE experiments, analysed the CE data, performed contact angle measurements and wrote the paper. M.O. and C.M.P. performed the LEIS measurements and analysed the corresponding data. M.H. carried out the ToF-SIMS measurements and M.H., F.F. and M.S. analysed the corresponding data. F.P. carried out the AFM measurements and analysed the corresponding data. D.M.B. carried out the XRD measurements and analysed the corresponding data. M.M.L., S.H.K. and D.L.S. contributed to the manuscript. A.B., F.G. and N.M. performed the FTIR measurements and analysed the corresponding data. S.R.W. conceived the project and experimental approach, secured funding and wrote the paper.

**Funding** Open access funding provided by Institute of Science and Technology (IST Austria).

**Competing interests** The authors declare no competing interests.

**Additional information**
**Correspondence and requests for materials** should be addressed to Galien Grosjean or Scott R. Waitukaitis.

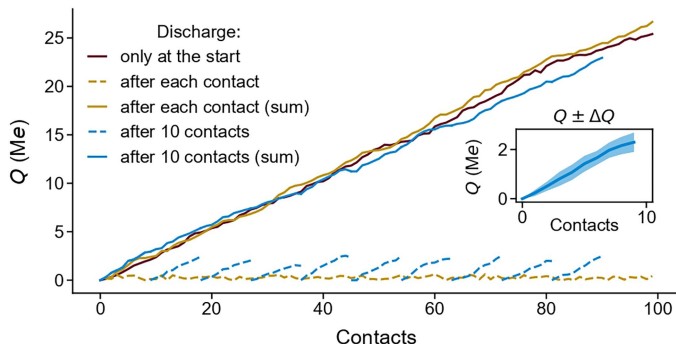

**Extended Data Fig. 1 | Photoionization protocol.** Charge measurements are performed in three different ways for the same pair of SiO₂ samples. First, 100 consecutive collisions/charge measurements are performed with a single 60-s discharge at the start. Second, 100 single-contact measurements are performed, in which the system is discharged after only one collision/charge measurement, so charge is not allowed to accumulate further. Finally, an intermediate measurement is performed, in which the system is discharged after each segment of ten charge measurements, repeated ten times for a total of 100 measurements. With the last of these, we can obtain the average charging behaviour and standard deviation (inset plot), which is the approach that was used in the main paper (with 10 collisions/11 measurements per segment instead). By summing individual measurements/segments, we confirm that the charging rates are identical in all three cases.

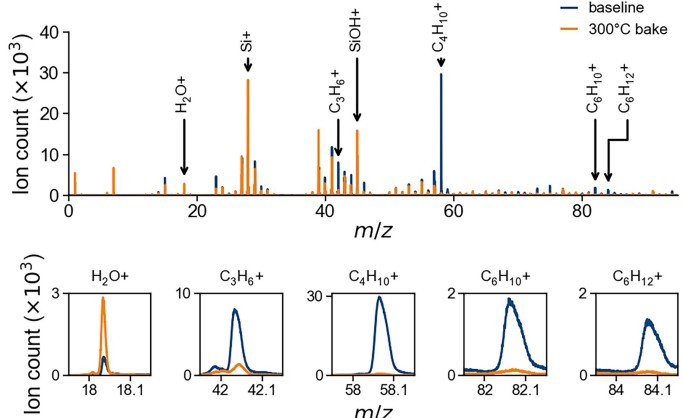

**Extended Data Fig. 2 | ToF-SIMS measurement before and after bake.**
ToF-SIMS spectra are taken before and after baking the sample at 300 °C for
16 h. The peaks corresponding to adventitious carbon are greatly reduced,
whereas peaks corresponding to the $SiO_2$ matrix, and other peaks owing to,
for example, alkali metals, are present both before and after the bake.

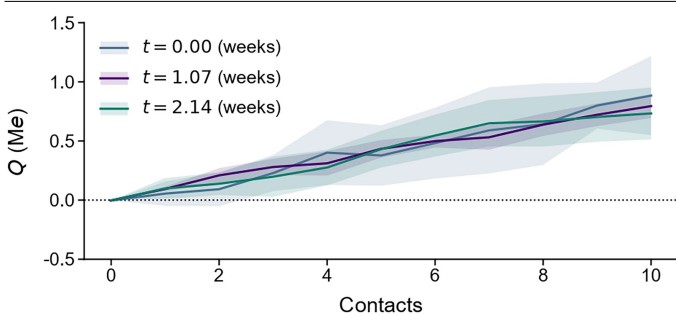

**Extended Data Fig. 3 | Stability of charging rates after standard cleaning protocol.** The charging rate of a SiO$_2$ pair is measured three times, with roughly a week between each measurement. The particle is kept in the acoustic trap at all times, in a closed box kept at 30% relative humidity. The solid line shows the average over ten series of ten collisions and the coloured region shows the corresponding standard deviation. We note that the 72-h waiting step in our protocol is required for this stability.

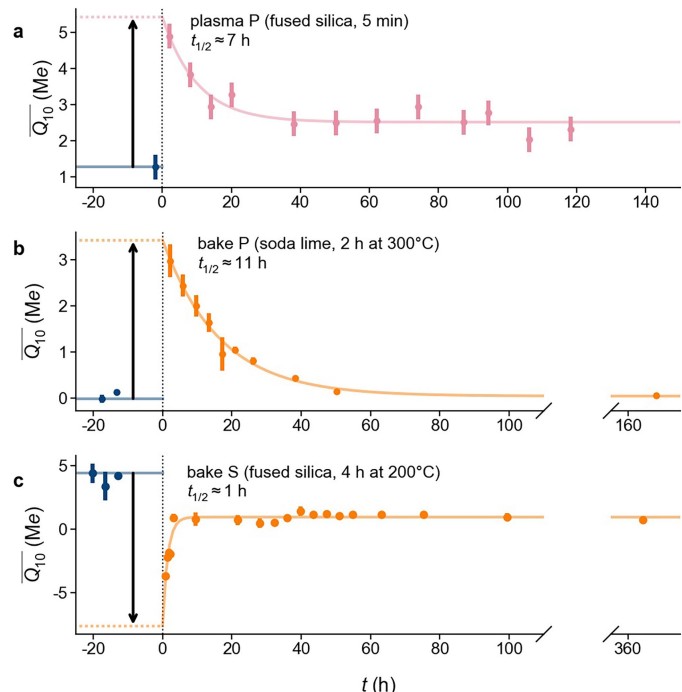

**Extended Data Fig. 4 | Charge relaxation after bake/plasma.** Three experiments show the evolution of the average ten-collision $\overline{Q_{10}}$ charging rate after bake or plasma. **a**, A $SiO_2$ sphere is measured against a $SiO_2$ plate. The plate is exposed to a plasma for 5 min at $t = 0$. **b**, A soda–lime glass sphere is measured against a soda–lime glass plate. The plate is baked at 300 °C for 2 h and taken out of the oven at $t = 0$. **c**, A $SiO_2$ sphere is measured against a $SiO_2$ plate. The sphere is baked at 200 °C for 4 h and taken out of the oven at $t = 0$. We find that the relaxation timescale ranges from approximately 1 to 10 h. Note that the timescale is not necessarily material specific; for example, here panels **a** and **c** both correspond to $SiO_2$, yet the timescale varies by about a factor of 10. Rather, this variability seems to correspond to fluctuating rates of carbon readsorption, probably because of fluctuating amounts of carbon impurities in the laboratory air.

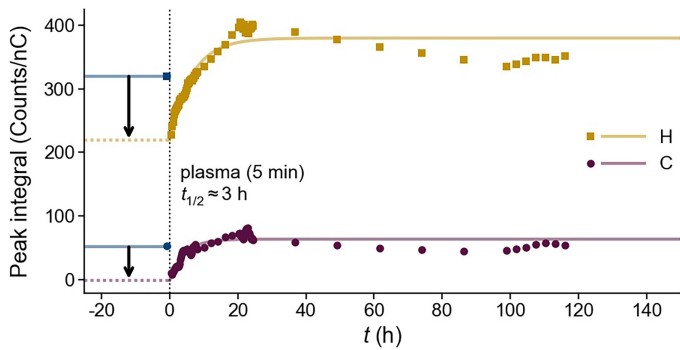

**Extended Data Fig. 5 | Time-of-flight H signal.** A signal for H can be extracted indirectly from the LEIS measurements using the time-of-flight filter. We compare it here with the corresponding LEIS signal for C, showing that both have a similar evolution in time. The fact that H does not go to zero can be attributed to silanol groups (Si−OH) and the quick readsorption of surface water.

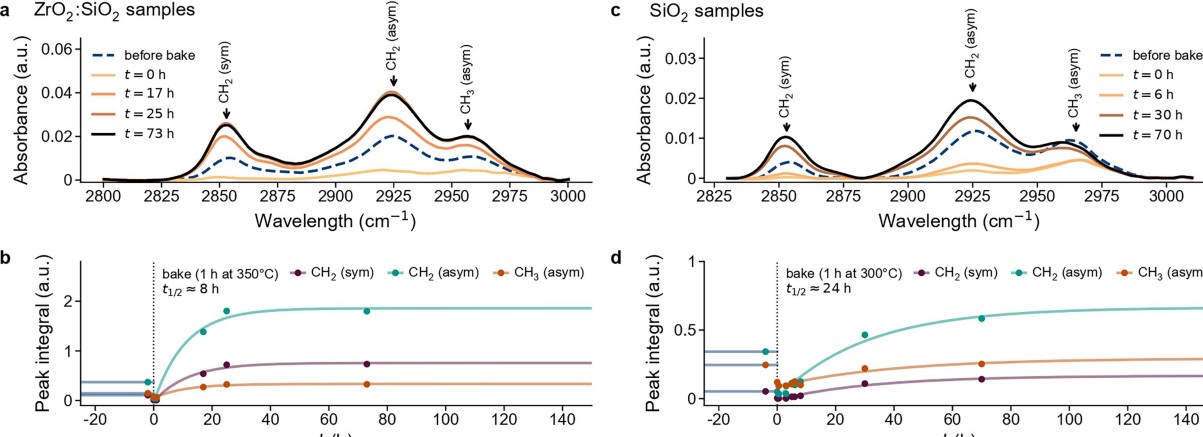

**Extended Data Fig. 6 | Diffuse reflectance infrared Fourier transform spectroscopy. a**, Vibrations of $CH_x$ species on diffuse reflectance infrared Fourier transform spectroscopy spectra reveal the presence of carbon on a 'clean' $ZrO_2$:$SiO_2$ sample. These signals decrease to zero immediately after baking and then progressively come back. **b**, By measuring the integrated absorbance peaks over time, we obtain a typical timescale $t_{1/2}$ of about 8 h for all species. **c**, On pure $SiO_2$ samples, we observe very similar dynamics. **d**, Here the timescale was longer, at $t_{1/2} \approx 24$ h.

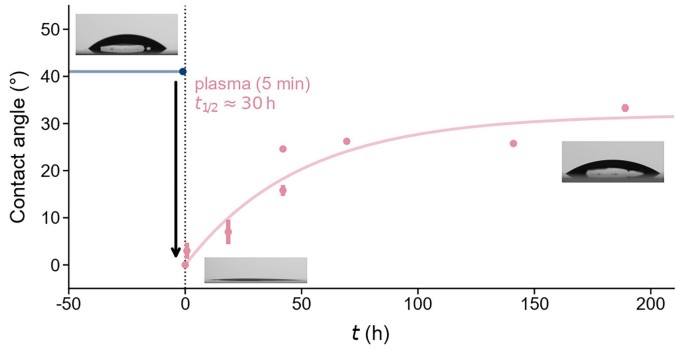

**Extended Data Fig. 7 | Contact angle evolution.** The contact angle of water on a fused silica slide (UQG Optics Ltd. FSM-7521) is measured with a contact angle goniometer (Ossila L2004A1). The slide is cleaned using the standard cleaning protocol described in Methods. At $t = 0$, it is exposed to a plasma for 5 min. For each measurement, a 10-µl droplet of ultrapure water is deposited with a precision glass syringe. Every measurement is performed at a different location on the plate. Between measurements, the plate is stored in the laboratory environment at around 40% relative humidity.

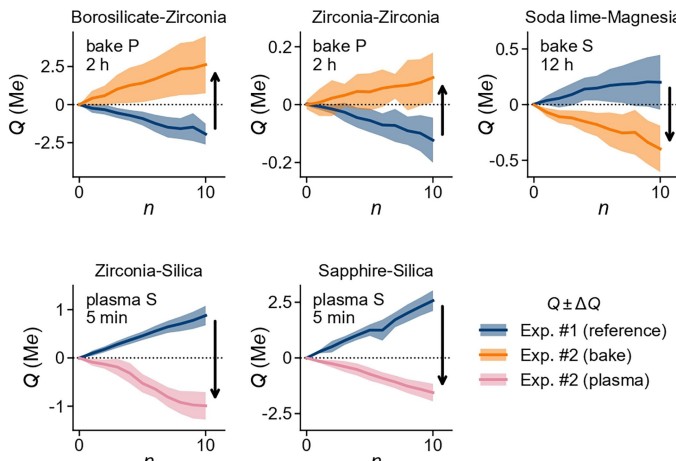

**Extended Data Fig. 8 | Controlling CE by removing carbon with other oxide materials.** Here we show five examples of charging polarity reversals as well as those shown in Fig. 4. This includes three examples in which the reversal was obtained by baking one of the samples at 300 °C: a borosilicate sphere contacted with a zirconia plate, a zirconia sphere contacted with a zirconia plate and a soda–lime sphere contacted with a magnesia plate. We also include two examples in which plasma treatment was used instead: a zirconia sphere contacted with a silica plate and a sapphire sphere contacted with a silica plate.

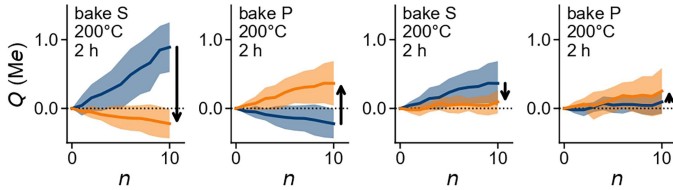

**Extended Data Fig. 9 | Near-complete suppression of charging by iterative carbon removal.** CE between the same pair of $SiO_2$ samples is successively measured, in which we alternatively bake the sphere or the plate for 2 h at 200 °C. The sample that has been baked last tends to charge more negatively. After several successive bakes, and hence near-complete carbon removal on both samples, the charging rate essentially goes to zero.

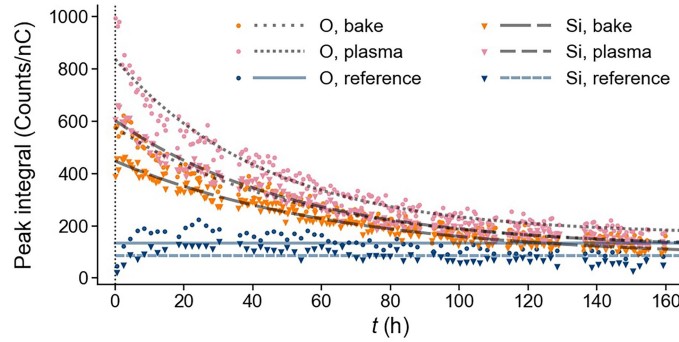

**Extended Data Fig. 10 | Readsorption of carbon in UHV.** The LEIS signals corresponding to the Si and O peaks at the surface of three samples stored in UHV are shown over time. One sample was exposed to plasma for 5 min, another was baked at 200 °C for 2 h and one was left untouched and kept as a reference. Immediately after bake/plasma, the Si and O signals increase considerably as the adsorbates covering the $SiO_2$ matrix are removed. This effect relaxes over a timescale of days as adsorbates come back to the surface. Hence we confirm as is widely to UHV practitioners–even in the cleanest vacuum–carbon comes back.