## [Peer Review File · Nature]

Adventitious carbon breaks symmetry in oxide contact electrification

Corresponding Author: Dr Galien Grosjean

Version 0:

Reviewer comments:

Referee #1

(Remarks to the Author)

Contact electrification is an effect everyone is aware of... it occurs when you rub a balloon on your hair and the balloon and hair get charged, and when you walk across a rug and get a shock as you go to touch a doorknob. It has enormous consequences in both industrial and natural settings. In industrial settings, discharges from charges that have built up from contact electrification can ignite explosions of combustible vapors or dust. In natural settings, such discharges are believed to have provided the activation energy for simple gases like methane to react to form amino acids that gave rise to life on earth. These are just two examples of the consequences of contact electrification... there are many, many more.

Even though contact electrification is ubiquitous and very consequential, the underlying physics is not understood. For example, there is no predictive theory to determine which surface charges positive or negative when two surfaces contact -- you just have to try it experimentally and see the result (there are "triboelectric series" that are used to anticipate which surface charges positive/negative, but these are merely catalogs of previous experimental results, and not always reproducible). I believe a key reason for this is that multiple processes may be in play, with different processes dominating in different situations.

This is the context in which the present paper comes in. Here, the authors show very convincingly that "adventitious carbon" plays an important -- and sometimes dominant -- role in contact electrification. This idea is totally novel. The idea that adsorbates in general can play a role in contact electrification is not new. However, as far as I'm aware, the only adsorbates identified as possible systematic drivers of contact electrification are water and ions that may be dissolved in the adsorbed water layer (e.g., OH⁻).

Even though adventitious carbon has not been discussed in the context of contact electrification (as far as I'm aware), it is well known -- particularly in the context of surface analyses like XPS -- that it is ubiquitous. In fact, adventitious carbon is frequently used for calibrating spectra in XPS. As one pertinent example that demonstrates both the ubiquity of adventitious carbon and no prior connection of adventitious carbon to contact electrification, a recent article [Liebauer et al JACS (2024)] explains their contact electrification results in terms of ions dissolved in water; these authors use XPS in their experiments and explicitly mention the use of adventitious carbon to calibrate their spectra, but the possible role of adventitious carbon in the contact electrification is never mentioned.

The present paper very convincingly demonstrates the role of adventitious carbon in contact electrification. First, I note that contact electrification experiments are very difficult, as any contact of the samples with other surfaces can lead to charge transfer, either by contact electrification as the contact occurs or by providing a path for charge to be transferred to/from ground for existing contacts. To overcome this issue, the authors have developed an elaborate and unique setup to acoustically levitate a particle so that it has no contact with other surfaces, and then manipulate the acoustic fields to force well controlled charging collisions of that particle with a target surface. The experimental setup also includes a way to the measure of charge of the particle without the particle contacting any extraneous surfaces. Finally, these experiments are carried in a very controlled environment.

It is with this unique experimental setup that the authors have obtained very high quality contact charging results. They then show that the charging behavior can be systematically and predictably altered by treatments that are believed to affect adsorbates on the surface -- in particular, heating and plasma treatments. The authors show that these treatments systematically and predictably cause a surface to charge more negatively.

To understand the reason that these treatments cause the surfaces to charge more negatively, the authors use a number of surface analyses techniques. These results show that the heating and plasma treatments are causing the removal of the

adventitious carbon that is on the surface under natural conditions.

The really compelling evidence is the time dependence results presented. After the surface treatments that remove the adventitious carbon, the surfaces will return in time to their natural states with adventitious carbon. And, as predicted by the authors, the contact charging changes caused by the surface treatment are reversed. Through a series of difficult experiments that measure both the time dependence of the contact charging behavior and the time dependence of the return of adventitious carbon, the authors show that the time scales for both effects are the same, and thus very convincingly demonstrate the dominant role of adventitious carbon in the contact electrification process.

This effects are demonstrated for a range of oxide materials, and is relevant for both contact between same-type materials (eg, between two SiO₂ surfaces) and dissimilar materials (eg, between SiO₂ and Al₂O₃). It is logical to think that similar effects of adventitious carbon will be relevant in all materials, not just oxide.

The paper is very clear and well written. The experiments and the data are robust. I like that the authors discussed some possibilities of how adventitious carbon may affect contact electrification, without pushing ideas that could not be gleaned from their experiments.

I recommend publishing the paper as is.

Daniel Lacks

Referee #2

(Remarks to the Author)

Extremely insightful and well-designed experiments. Interruption of an acoustic field and high speed imaging of the trajectory of a sphere under an oscillating field. Technically sound and a good addition to the practical knowledge on static electrification of insulators (silica, alumina and zirconia). I do not believe it will seal off the debate on why identical materials "can" gain opposite charging (magnitude and sign). Still, it will be received well by the community.

The main conclusion is that the charging behaviour of insulating oxides correlates with the presence of carbon-based contaminations. I do not believe that the use of "control" in the title is appropriate because it conveys the idea that mechanical factors and surface deformation factors would be secondary to contaminations. I do not think there is evidence to suggest this.

Regarding the role of contaminants. Researchers in the field have obviously known this for a long time: there were speculations, but also direct evidence. Surfaces do gain charge upon contact when perfectly clean (UHV experiments done in the 80s, K.P. Homewood, J. Electrostatics, 10, 1981, 299-304) but "extrinsic" surface states definitely play a role (e.g. J. Chem. Phys. 61, 1455-1462 (1974)). Relevant to the findings of this paper, the adsorption of carbonaceous (~1nm thin) monolayers dramatically changes the static charging of silicon (AFM experiments in Nano Energy, 2022, 93, 106861). Surface states are the "storage" and is nice to see that evidence towards this is growing.

The statement "symmetry-breaking parameter that causes this remains unidentified" is a stretch to far. Perhaps we do not yet have the complete picture, but asymmetry in surface curvature (from the macro to the nanoscale) has been repeatedly linked to statics (or at least to the driving force behind charging; Physical Review Letters 123, 116103 (2019)). Under most circumstances, polarisation in response to strain gradients explains why even identical insulating materials gain opposite charges upon contact. This was first shown by Jameson, a school teacher from Glasgow, and published in a 1910 Nature paper (DOI: 10.1038/083189a0; concave versus convex sheets of celluloid) and much more recently by Burgo (T.A.L. Burgo, A. Erdemir. Angew. Chem. Int. Ed. 53 (2014), 12101; tribocharging tracking gradients reverse from indent to pullouts), by Z. L. Wang (C. Xu et al., ACS Nano. 2019, 13, 2034; sign of charge/potential depends upon the curvature when using two pieces of the same material), by B. Persson (Persson, B. N. J. . EPL (Europhysics Letters, 2020, 129, 10006; local curvature differences, indenter versus indente) and by Jaeger (S.R. Waitukaitis, et al. Phys. Rev. Lett. 112 (2014), 218001; contact between fused zirconium dioxide silicate particles of different sizes with the size (curvature) leading to asymmetry of charging. Claiming that the symmetry breaker are asymmetric impurities is surely the case for the experiments reported here, but in many other contact electrification systems the explanation is curvature (direction of the strain gradient).

Geometric curvature is likely to be the main symmetry breaker, and even if instantaneous flexoelectric fields are symmetric, their relaxation dynamics can be asymmetric, perhaps due to adsorbed impurities, and this I believe should be the main claim of this report.

Referee #3

(Remarks to the Author)

This manuscript discusses an experimental investigation that strongly points towards adventitious carbon as the determining factor in triboelectric charging between insulating oxides. Overall, I found this to be a very convincing paper. It is relevant and appropriate for publication Nature because it is relevant to a wide audience (tribocharging is ubiquitous) and is a significant improvement in humanity's understanding of this phenomenon. This is, to my knowledge, the first rigorous demonstration that adventitious carbon is a significant factor in tribocharging of insulators. Prior work has hypothesized charge transfer of ions, electrons, or water. This work is a significant step forward.

Methods: The paper uses acoustic levitation to suspend and then bounce a single bead, and then measure the charge on

the bead. This method has been pioneered by the senior author in prior works. It is an appropriate method for the investigation because it allows the measurement of charging after a single bounce and the tight control of the sample and atmospheric conditions. I have no concerns regarding the use of statistics in this work.

Conclusions: The conclusions are supported by the data. The conclusions are properly qualified and do not extend beyond their region of relevance.

I have a few minor comments that I think would improve the paper.

1. Line 175: why does the charging rate find a new stable value? Is this just due to the atmospheric conditions that the sample is exposed to?
2. Line 215: The reference to Fig 3C does not seem relevant here.
3. Line 251-252: "If the carbon symmetry is minimal, material properties win. If the one surface is largely stripped..." I felt like I was following the rest of the paper, and then these two sentences confused me. My initial thought after the first sentence was "if the surfaces are covered in carbon, isn't the charge transfer driven by carbon transfer?" Is the "material property" just "how tightly the material bonds to hydrocarbons"? I think a few more words here to more explicitly describe the thought would be useful.

Referees' comments:

Individual remarks are addressed point-by-point below. The reports from the Referees are written in *blue italics* and our responses in black upright text.

Referee #1 (Remarks to the Author):

Contact electrification is an effect everyone is aware of... it occurs when you rub a balloon on your hair and the balloon and hair get charged, and when you walk across a rug and get a shock as you go to touch a doorknob. It has enormous consequences in both industrial and natural settings. In industrial settings, discharges from charges that have built up from contact electrification can ignite explosions of combustible vapors or dust. In natural settings, such discharges are believed to have provided the activation energy for simple gases like methane to react to form amino acids that gave rise to life on earth. These are just two examples of the consequences of contact electrification... there are many, many more.

Even though contact electrification is ubiquitous and very consequential, the underlying physics is not understood. For example, there is no predictive theory to determine which surface charges positive or negative when two surfaces contact -- you just have to try it experimentally and see the result (there are "triboelectric series" that are used to anticipate which surface charges positive/negative, but these are merely catalogs of previous experimental results, and not always reproducible). I believe a key reason for this is that multiple processes may be in play, with different processes dominating in different situations.

This is the context in which the present paper comes in. Here, the authors show very convincingly that "adventitious carbon" plays an important -- and sometimes dominant -- role in contact electrification. This idea is totally novel. The idea that adsorbates in general can play a role in contact electrification is not new. However, as far as I'm aware, the only adsorbates identified as possible systematic drivers of contact electrification are water and ions that may be dissolved in the adsorbed water layer (e.g., OH-).

Even though adventitious carbon has not be discussed in the context of contact electrification (as far as I'm aware), it is well known -- particularly in the context of surface analyses like XPS -- that it is ubiquitous. In fact, adventitious carbon is frequently used for calibrating spectra in XPS. As one pertinent example that demonstrates both the ubiquity of adventitious carbon and no prior connection of adventitious carbon to contact electrification, a recent article [Liebauer et al JACS (2024)] explains their contact electrification results in terms of ions dissolved in water; these authors use XPS in their experiments and explicitly mention the use of adventitious carbon to calibrate their spectra, but the possible role of adventitious carbon in the contact electrification is never mentioned.

The present paper very convincingly demonstrates the role of adventitious carbon in contact electrification. First, I note that contact electrification experiments are very difficult, as any contact of the samples with other surfaces can lead to charge transfer, either by contact electrification as the contact occurs or by providing a path for charge to be transferred to/from ground for existing contacts. To overcome this issue, the authors have developed an elaborate and unique setup to acoustically levitate a particle so that it has no contact with other surfaces, and then manipulate the acoustic fields to force well controlled charging collisions of that particle with a target surface. The experimental setup also includes a way to the measure of charge of the particle without the particle contacting any extraneous surfaces. Finally, these experiments are carried in a very controlled environment.

It is with this unique experimental setup that the authors have obtained very high quality contact charging results. They then show that the charging behavior can be systematically and predictably altered by treatments that are believed to affect adsorbates on the surface -- in particular, heating and plasma treatments. The authors show that these treatments systematically and predictably cause a surface to charge more negatively.

To understand the reason that these treatments cause the surfaces to charge more negatively, the authors use a number of surface analyses techniques. These results show that the heating and plasma treatments are causing the removal of the adventitious carbon that is on the surface under natural conditions.

The really compelling evidence is the time dependence results presented. After the surface treatments that remove the adventitious carbon, the surfaces will return in time to their natural states with adventitious carbon. And, as predicted by the authors, the contact charging changes caused by the surface treatment are reversed. Through a series of difficult experiments that measure both the time dependence of the contact charging behavior and the time dependence of the return of adventitious carbon, the authors show that the time scales for both effects are the same, and thus very convincingly demonstrate the dominant role of adventitious carbon in the contact electrification process.

This effects are demonstrated for a range of oxide materials, and is relevant for both contact between same-type materials (eg, between two SiO₂ surfaces) and dissimilar materials (eg, between SiO₂ and Al₂O₃). It is logical to think that similar effects of adventitious carbon will be relevant in all materials, not just oxide.

The paper is very clear and well written. The experiments and the data are robust. I like that the authors discussed some possibilities of how adventitious carbon may affect contact electrification, without pushing ideas that could not be gleaned from their experiments.

I recommend publishing the paper as is.

Daniel Lacks

We are extremely grateful for Prof. Lacks' enthusiastic endorsement and beautiful description of our results.

Referee #2 (Remarks to the Author):

Extremely insightful and well-designed experiments. Interruption of an acoustic field and high speed imaging of the trajectory of a sphere under an oscillating field. Technically sound and a good addition to the practical knowledge on static electrification of insulators (silica, alumina and zirconia). I do not believe it will seal off the debate on why identical materials "can" gain opposite charging (magnitude and sign). Still, it will be received well by the community.

We thank Referee #2 very much for their kind words.

The main conclusion is that the charging behaviour of insulating oxides correlates with the presence of carbon-based contaminations. I do not believe that the use of "control" in the title is appropriate because it conveys the idea that mechanical factors and surface deformation factors would be secondary to contaminations. I do not think there is evidence to suggest this.

We agree with Referee #2's assessment that "control" is perhaps not the best choice of words to describe our results. In the revised version, we have changed the title to better highlight the main result of the paper, which is that the difference in adsorbed carbon is the symmetry-breaking parameter that explains why same-material oxide surfaces can exchange charge continually. While we believe that mechanical/deformation factors are important things to consider in general, we are also confident that our results show that they are not, at least in the oxide contacts we have studied, the source of symmetry breaking. For instance, the carbon effect is still present even when samples differ in roughness by multiple orders of magnitude (see Fig. 4a), and is observed between spherical and flat samples throughout the paper. This is discussed in more detail below.

Regarding the role of contaminants. Researchers in the field have obviously known this for a long time: there were speculations, but also direct evidence. Surfaces do gain charge upon contact when perfectly clean (UHV experiments done in the 80s, K.P. Homewood, J. Electrostatics, 10, 1981, 299-304) but "extrinsic" surface states definitely play a role (e.g. J. Chem. Phys. 61, 1455–1462 (1974)). Relevant to the findings of this paper, the adsorption of carbonaceous (~1nm thin) monolayers dramatically changes the static charging of silicon (AFM experiments in Nano Energy, 2022, 93, 106861). Surface states are the "storage" and is nice to see that evidence towards this is growing.

We agree with Referee #2's assessment: our results join a long list of evidence showing that surface states play an important role in tribocharging. We discuss two such references with relevance to oxides in the manuscript.

The statement “symmetry-breaking parameter that causes this remains unidentified” is a stretch to far. Perhaps we do not yet have the complete picture, but asymmetry in surface curvature (from the macro to the nanoscale) has been repeatedly linked to statics (or at least to the driving force behind charging; Physical Review Letters 123, 116103 (2019)). Under most circumstances, polarisation in response to strain gradients explains why even identical insulating materials gain opposite charges upon contact. This was first shown by Jameson, a school teacher from Glasgow, and published in a 1910 Nature paper (DOI: 10.1038/083189a0; concave versus convex sheets of celluloid) and much more recently by Burgo (T.A.L. Burgo, A. Erdemir. Angew. Chem. Int. Ed. 53 (2014), 12101; tribocharging tracking gradients reverse from indent to pullouts), by Z. L. Wang (C. Xu et al., ACS Nano. 2019, 13, 2034; sign of charge/potential depends upon the curvature when using two pieces of the same material), by B. Persson (Persson, B. N. J. . EPL (Europhysics Letters, 2020, 129, 10006; local curvature differences, indenter versus indente) and by Jaeger (S.R. Waitukaitis, et al. Phys. Rev. Lett. 112 (2014), 218001; contact between fused zirconium dioxide silicate particles of different sizes with the size (curvature) leading to asymmetry of charging. Claiming that the symmetry breaker are asymmetric impurities is surely the case for the experiments reported here, but in many other contact electrification systems the explanation is curvature (direction of the strain gradient).

Geometric curvature is likely to be the main symmetry breaker, and even if instantaneous flexoelectric fields are symmetric, their relaxation dynamics can be asymmetric, perhaps due to adsorbed impurities, and this I believe should be the main claim of this report.

While we agree with Referee #2 that, for certain materials, geometric or mechanical parameters seem to be key, our data does not support the assertion that they play a role—even an indirect role—in our experiments. We must reiterate that (1) for all the oxides we have tried, spherical and flat samples were equally likely to charge positive/negative, as seen for instance in Fig. 1e for SiO₂, and (2) charge reversal occurred consistently between a sphere and a plate, between a rough surface and a smooth surface, etc., as shown e.g. in Fig. 4. While our findings might still be reconciled with a purely flexoelectricity-based mechanism by invoking some effect of adsorbates (e.g., on the flexoelectric coefficient), this is speculative and should not be ‘the main claim of this report’. Out of prudence, we do not rule it out entirely.

We appreciate Referee #2's extensive knowledge of the literature on contact electrification broadly. However, we must note that much of the work they point to does not concern oxides, with one exception that we address now: the free-fall experiment in Waitukaitis et al., PRL **112**, 218001 (2014).

In the free-fall paper, small and large oxide grains were shown to charge negative and positive, respectively. Importantly, charge was measured after an unknown (presumably very large) number of collisions in a grain hopper. Without access to the full charging history of individual grains, one can only extrapolate to the origins of the symmetry breaking. While it is reasonable to assume that size-dependent charging might have occurred throughout, we point to theoretical work which showed that it can emerge after many collisions even if the charging mechanism is initially independent on size, see e.g. Preud'homme *et al.*, *Soft Matter* **19**, 8911 (2023), or Grosjean & Waitukaitis, *Phys. Rev. Materials* **7**, 065601 (2023). This arises naturally if we assume that there is a finite number of charge donating/receiving sites, which tend to deplete faster on smaller grains as they experience more collisions per surface area. While size-dependent charging is not yet fully understood, this shows it is at least not incompatible with our findings.

We do not claim nor would we defend the idea that there is one mechanism for all material classes. This is why we cautiously restricted our claim to oxides. In the revised version of the paper, we rephrase our mention of flexoelectricity in the conclusion to make our position clearer to the reader.

Referee #3 (Remarks to the Author):

This manuscript discusses an experimental investigation that strongly points towards adventitious carbon as the determining factor in triboelectric charging between insulating oxides. Overall, I found this to be a very convincing paper. It is relevant and appropriate for publication Nature because it is relevant to a wide audience (tribocharging is ubiquitous) and is a significant improvement in humanity's understanding of this phenomenon. This is, to my knowledge, the first rigorous demonstration that adventitious carbon is a significant factor in tribocharging of insulators. Prior work has hypothesized charge transfer of ions, electrons, or water. This work is a significant step forward.

Methods: The paper uses acoustic levitation to suspend and then bounce a single bead, and then measure the charge on the bead. This method has been pioneered by the senior author in prior works. It is an appropriate method for the investigation because it allows the measurement of charging after a single bounce and the tight control of the sample and atmospheric conditions. I have no concerns regarding the use of statistics in this work.

Conclusions: The conclusions are supported by the data. The conclusions are properly qualified and do not extend beyond their region of relevance.

We thank Referee #3 very much for their kind words. Individual comments are addressed point-by-point below.

I have a few minor comments that I think would improve the paper.
1. Line 175: why does the charging rate find a new stable value? Is this just due to the atmospheric conditions that the sample is exposed to?

It is difficult to say exactly what causes the stable values before/after treatment to differ, but our guess is that it is indeed due to atmospheric conditions. More precisely, we believe that if a broad range of carbon species are present in trace amounts, we can expect some variability as to precisely what gets on the surface and in what quantity. LEIS and FTIR measurements show a similar behavior, which indicates that the composition of adsorbates must vary.

2. Line 215: The reference to Fig 3C does not seem relevant here.

Indeed, Fig. 3c on its own does not show multiple timescales. We meant to include it as an extra example to compare it with the three shown in Extended Data Fig. 4. We agree that this is perhaps unclear and have removed the reference in the revised version.

3. Line 251-252: “If the carbon symmetry is minimal, material properties win. If the one surface is largely stripped...” I felt like I was following the rest of the paper, and then these two sentences confused me. My initial thought after the first sentence was “if the surfaces are covered in carbon, isn’t the charge transfer driven by carbon transfer?” Is the “material property” just “how tightly the material bonds to hydrocarbons”? I think a few more words here to more explicitly describe the thought would be useful.

In the revised version, we have modified these lines to improve clarity. What we mean by the ‘intermediate’ scenario is that both carbon and the underlying material seem to matter, but which one wins depends on the conditions. If both surfaces are covered in carbon, they behave like different materials as one would expect. But if we break that balance by stripping one surface from its carbon, we can cause charge to go the ‘wrong’ way (as predicted by the triboelectric series). How precisely these two contributions interact, we cannot say for now. Maybe different materials tend to accumulate different amounts/kinds of carbon; maybe the way the carbon binds to the surface is important for charge transfer; or maybe the carbon modifies some other property of the underlying material (e.g. work function). Our hope is that, now that we know to focus on adventitious carbon, we might be able to answer such questions in the not-so-distant future.